# The hormonal peptide Elabela guides angioblasts to the midline during vasculogenesis

Christian SM Helker[1][*][†], Annika Schuermann[1][†], Cathrin Pollmann[2], Serene C Chng[3], Friedemann Kiefer[2,4], Bruno Reversade[3], Wiebke Herzog[1,2,4][*]

[1]University of Muenster, Muenster, Germany; [2]Max Planck Institute for Molecular Biomedicine, Muenster, Germany; [3]Institute of Medical Biology, Human Genetics and Embryology Laboratory, A*STAR, Singapore, Singapore; [4]Cells-in-Motion Cluster of Excellence, University of Muenster, Muenster, Germany

**Abstract** A key step in the de novo formation of the embryonic vasculature is the migration of endothelial precursors, the angioblasts, to the position of the future vessels. To form the first axial vessels, angioblasts migrate towards the midline and coalesce underneath the notochord. Vascular endothelial growth factor has been proposed to serve as a chemoattractant for the angioblasts and to regulate this medial migration. Here we challenge this model and instead demonstrate that angioblasts rely on their intrinsic expression of Apelin receptors (Aplr, APJ) for their migration to the midline. We further show that during this angioblast migration Apelin receptor signaling is mainly triggered by the recently discovered ligand Elabela (Ela). As neither of the ligands Ela or Apelin (Apln) nor their receptors have previously been implicated in regulating angioblast migration, we hereby provide a novel mechanism for regulating vasculogenesis, with direct relevance to physiological and pathological angiogenesis.

*For correspondence: Christian.
Helker@mpi-bn.mpg.de (CSMH);
wiebke.herzog@mpi-muenster.
mpg.de (WH)

†These authors contributed
equally to this work

Reviewing editor: Tanya T
Whitfield, University of Sheffield,
United Kingdom

## Main text

In the vertebrate embryo, the formation of the large axial vessels, namely the dorsal aorta (DA) and the cardinal vein (CV), establishes a first circulatory loop and thereby the core of the developing cardiovascular system. Angioblasts are initially specified in the lateral plate mesoderm and migrate between the somites towards the midline, where they coalesce and assemble the DA and the CV underneath the notochord (NC) (*Figure 1A,B–F*, *Video 1*).

It has been previously proposed that this process is regulated by Vascular endothelial growth factor A (VEGF-A) (*Coultas et al., 2005*; *Verma et al., 2010*; *Gore et al., 2012*), a master regulator of vascular growth in the embryonic and adult organism. In *Xenopus* embryos, transcripts for the *vegfa* gene are expressed in the midline and migrating angioblasts were guided towards sites of ectopic Vegfa expression (*Cleaver and Krieg, 1998*). Global inactivation of the genes encoding VEGF-A or the corresponding receptor VEGFR2 in mice led to strong vascular defects affecting the whole vascular network including the DA (*Shalaby et al., 1995*; *Ferrara et al., 1996*). Based on these data it was concluded that in vertebrates VEGF-A signaling regulates angioblast migration to the midline (*Coultas et al., 2005*; *Verma et al., 2010*; *Gore et al., 2012*). To get direct, mechanistic and dynamic insight into angioblast migration and formation of the great vessels, we decided to investigate these processes in zebrafish embryos.

As previously published (*Nicoli et al., 2008*), expression analysis of *vegfaa,* the gene for the main VEGF-A ortholog in zebrafish, showed the presence of transcripts in the somites between 12 and 15 hr post fertilization (hpf), that is the stage when angioblasts migrate to the midline (see *Figure 1B–F*). To

**eLife digest** The circulatory system enables blood to move around the body and deliver substances including nutrients and oxygen to the cells that need them. In the embryos of animals with a backbone, blood flows from the heart through the aorta into branching smaller vessels to the cells. The blood then gets collected by progressively bigger vessels and flows back to the heart via the cardinal vein. The cells that make up these blood vessels develop from cells called angioblasts—but first, during development these angioblasts must move to the place where the vessels will form.

A protein called Vascular endothelial growth factor (VEGF) had been suggested to help guide and align the angioblasts as the embryo develops. Now, Helker, Schuermann et al. have examined developing zebrafish embryos using new technologies. This revealed that VEGF is in fact not essential for the dorsal aorta and cardinal vein to develop. Instead, the angioblasts only move to the correct part of the embryo if they can produce the Apelin receptor protein, which forms part of a signaling pathway.

There are two hormones—called Apelin and Elabela—that can bind to and activate the Apelin receptor. Helker, Schuermann et al. show that Elabela alone is needed to guide the angioblasts to the right part of the embryo during blood vessel development. However, in embryos where there is not enough Elabela, the Apelin hormone can compensate for this deficiency and the first blood vessels will later develop correctly. Future research will address whether this signaling pathway not only guides angioblasts to establish a circulatory system, but also guides blood vessel growth. As blood vessel growth is very relevant to human disease, identifying the mechanisms that regulate it will have an impact on biomedical research.

analyze whether Vegfa signaling, as previously proposed, is required for angioblast migration to the midline, we analyzed loss-of-function zebrafish embryos with deficiencies in the Vegfa signaling pathway resulting from genomic mutations as well as morpholino (MO) -mediated knockdown. To rule out developmental delays, we counted the somites of the embryos and fixed them for confocal analysis, when they had developed 16–17 somites (equivalent to 17 hpf). Surprisingly, neither Vegfa ligand depletion (*vegfaa/vegfab* MO) nor a null mutation in the gene encoding the VEGFR2 ortholog (*kdrl*) interfered with angioblast migration in zebrafish embryos (*Figure 1H,J*), although Vegfa signaling dependent phenotypes, like failure to form the intersegmental vessels, could be observed after 24 hpf (*Figure 1—figure supplement 1*). Additionally, pharmacological inhibition of Vegfr2 signaling from 6 to 17 hpf did not impair angioblast migration to the midline (*Figure 1—figure supplement 1*). As activity of Sonic Hedgehog (Shh) in the midline has been shown to induce *vegfaa* expression (*Lawson et al., 2002*), we repeated the published experiments and blocked this process by chemical inhibition of Shh signaling using cyclopamine. In line with our previous experiments, angioblast migration was not affected after Shh inhibition (*Figure 1—figure supplement 2*). The sum of these data strongly indicates that Vegfa signaling is dispensable for angioblast specification or migration in zebrafish.

In order to identify the endogenous signal(s) guiding angioblast migration, we further analyzed the expression pattern of factors involved in cell motility and directed cell migration. As previously published (*Zeng et al., 2007*), we detected the expression of Apelin, a short secreted peptide encoded by the *apln* gene, by the NC (*Figure 2—figure supplement 1C*). At the same time, transcripts of its two paralogous receptors *apelin receptor a* (*aplnra*) and *apelin receptor b* (*aplnrb*) were present in angioblasts (*Figure 2—figure supplement 1A,B*; [*Scott et al., 2007*; *Zeng et al., 2007*]). Using MO-mediated gene knockdown, we observed that both *aplnra* and *aplnrb* are important for midline migration of angioblasts. While absence of each individual receptor reduced the efficiency of migration towards the midline, double depletion completely abolished this process (*Figure 2A*, *Video 2*). We used TALEN or CRISPR/Cas9-mediated gene editing (*Hwang et al., 2013*; *Jao et al., 2013*) to introduce mutations in either the *aplnra* or the *aplnrb* gene (*Figure 2—figure supplement 2*). As our MO-based analysis already indicated an additive effect of both receptor genes, we analyzed the phenotypes of the offspring of *aplnra*[+/−];*aplnrb*[+/−] double heterozygous parents. The embryos were all individually scored for their phenotype regarding angioblast midline

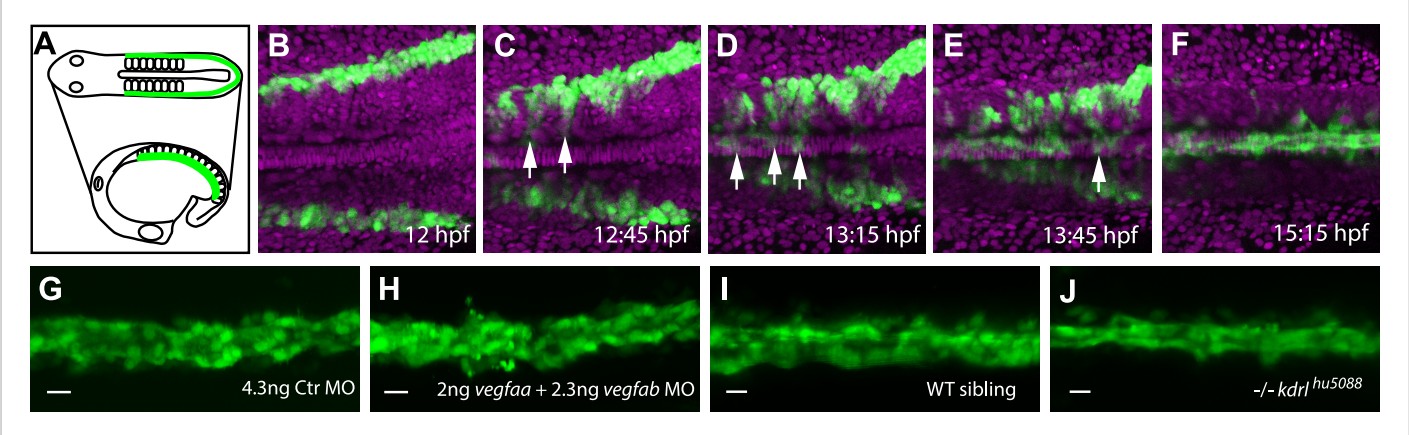

**Figure 1**. Angioblast migration to the midline is not regulated by Vegf. (**A–F**) Schematic (**A**) and confocal (**B–F**) in vivo time-lapse imaging of angioblasts (green) and their migration in a *Tg(fli1a:EGFP)^y1* embryo, injected with *H2B-mCherry* mRNA (purple, all nuclei), dorsal views at indicated time points. Arrows indicate the initial migration to the midline, and the coalescing into the dorsal aorta (DA). (**G–J**) Loss of Vegf-signaling components by *vegfaa/vegfab* double morpholino (MO) injection (**H**) or in *vegf receptor 2* (*kdrl*) mutants (**J**) does not affect angioblast migration. Confocal projections of *Tg(fli1a:EGFP)^y1* embryos in dorsal views at 17 hpf. For each condition, n > 15.

The following source data and figure supplements are available for figure 1:

**Source movie 1**. Time-lapse movie showing angioblast (green) migration to the midline analyzed using *Tg(fli1a:EGFP)^y1* embryos injected with *H2B-mCherry* mRNA (purple, all nuclei).

**Figure supplement 1**. Inhibition of Vegfa signaling resulted in normal angioblast migration to the midline, but impaired sprouting angiogenesis.

**Figure supplement 2**. Abrogation of Shh signaling abolishes *vegfaa* expression, but does not influence angioblast migration.

migration into four categories (normal, mild, strong, stronger; *Figure 2B*) and then subjected to genotyping. Homozygous mutant embryos phenocopied the MO-induced phenotypes, additional loss of one or two copies of the second receptor increased the severity of the phenotype (*Figure 2C*). Additionally, we detected *aplnrb* transcripts together with the angioblast specification marker *etv2* indicating that both genes are expressed very early during angioblast specification. However, when we analyzed *aplnra/aplnrb* double deficient embryos, no difference in *etv2* expression was detectable by in situ hybridization indicating that Apln receptors only regulate angioblast migration, but not specification (*Figure 2—figure supplement 1E*).

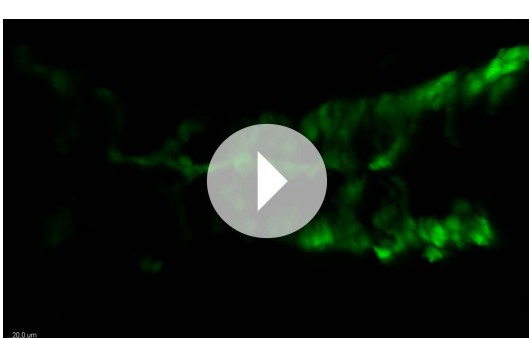

**Video 1.** Angioblast migration between 12.5 hpf and 16 hpf observed by time-lapse imaging using *Tg(fli1a: EGFP)^y1* to visualize angioblasts. Angioblasts migrate to the midline in Wt embryos.

Next, we analyzed the requirement for the ligand Apln by MO-mediated gene knockdown and, unexpectedly, observed no difference to control MO injected embryos (*Figure 2A*). To obtain undisputable genetic confirmation of this result, we again used gene editing to introduce mutations in the *apln* gene (*Figure 2—figure supplement 2*, most likely resulting in functional null mutants). Consistent with the MO-mediated depletion, homozygous *apln^mu267* mutant embryos (*Figure 2A*) showed no vasculogenesis defects, which indicated that endogenous Apln is not sufficient to regulate angioblast migration to the midline.

Previous studies have identified a requirement for Apln receptors in myocardial development in zebrafish (*Scott et al., 2007*; *Zeng et al., 2007*) and mice (*Charo et al., 2009*), which could not

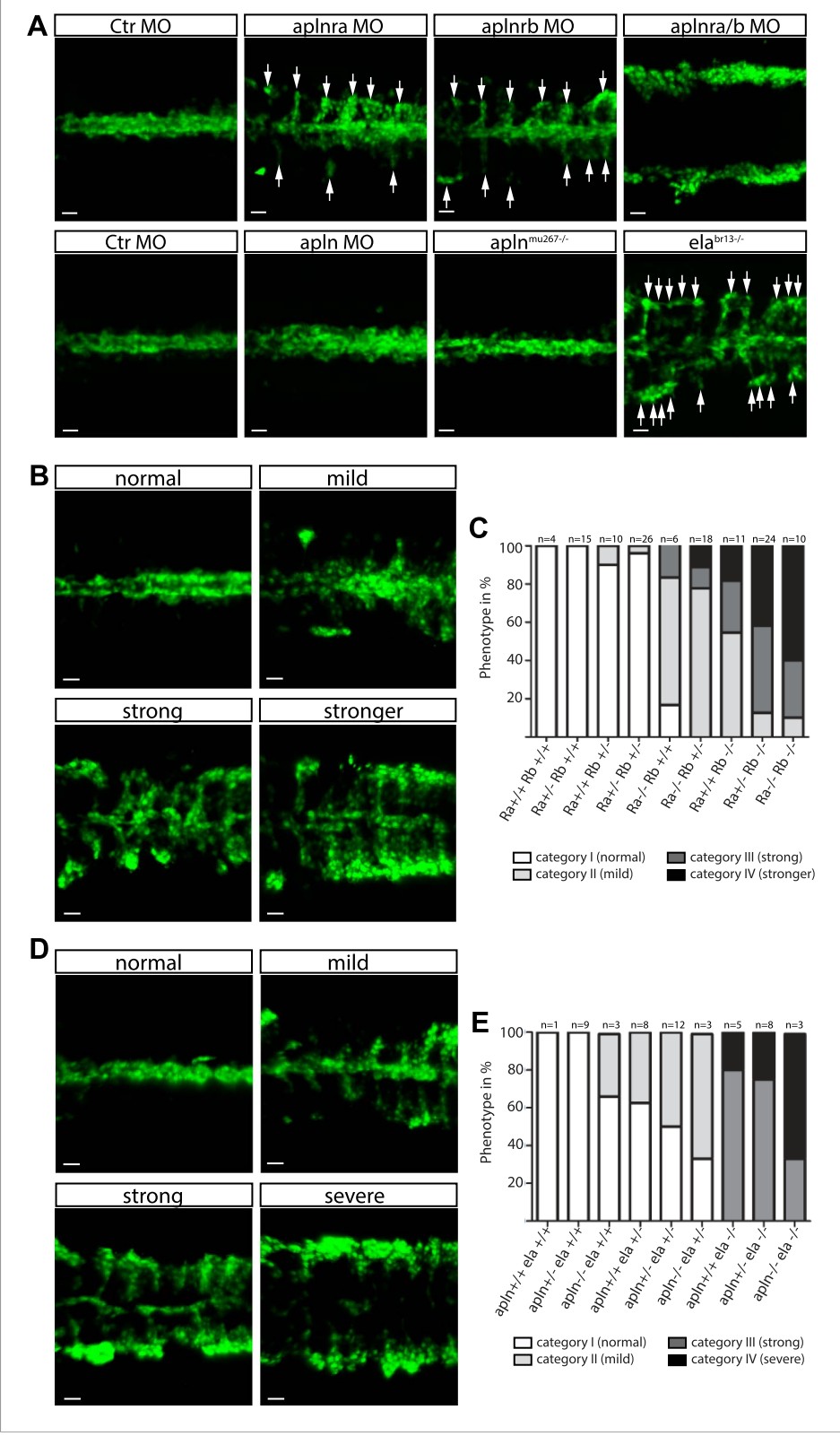

**Figure 2**. Ela/Apelin receptor - signaling guides angioblast migration to the midline. (**A**) Angioblasts have migrated to the midline at 17 hpf in wild type, *apln^{mu267}* mutant or *apln* MO injected zebrafish embryos. MO-mediated knockdown of *apelin receptor a* or *b* partially inhibited midline migration, while simultaneous loss of both *apelin receptor* genes completely abolished midline migration of angioblasts. Likewise, embryos with homozygous

*Figure 2. continued on next page*

*Figure 2. Continued*

mutations in the ligand *ela* display impaired migration of angioblasts. Arrows indicate aberrant positions.
(**B**, **C**) Mutations in *apln receptor* genes impair angioblast migration. Analysis of the offspring of *aplnra$^{+/-}$;aplnrb$^{+/-}$* double heterozygous parents resulted in four phenotypic categories (normal, mild, strong, stronger). (**B**) Genotyping of individual embryos revealed an additive effect, while mild phenotypes were observed when one receptor gene was homozygously mutant, phenotypic strength increased with additional loss of functional receptor genes (c; Ra, *aplnra*; Rb, *aplnrb*). (**D**, **E**) Ela deficiency can partially be compensated by Apln. Analysis of the offspring of *apln$^{+/-}$; ela$^{+/-}$* double heterozygous parents resulted in four phenotypic categories (normal, mild, strong, severe).
(**D**) Genotyping of individual embryos revealed a dose dependency, with increasing phenotypic strength correlating with additional loss of *apln* alleles in *ela* mutant embryos. (**E**) *ela$^{-/-}$; apln$^{-/-}$* double mutants phenocopied *apln receptor* deficiency. Angioblasts (green) were labeled by *Tg(fli1a:EGFP)$^{y1}$* expression, scale bars represent 30 μm.

The following source data and figure supplements are available for figure 2:

**Source data 1**. Excel table showing the phenotype categories and the number of embryos for each genotype in these phenotypic categories (for the *apln/ela* double mutant analysis).

**Figure supplement 1**. Expression of *aplnra*, *aplnrb*, *apln* and *ela* coincides with the formation of the DA and PCV.

**Figure supplement 2**. Generation of zebrafish *aplnra*, *aplnrb* and *apln* mutants.

completely be phenocopied by Apln deficiency (*Scott et al., 2007*; *Zeng et al., 2007*; *Charo et al., 2009*). These results led to the proposal that Apln may not be the only ligand for Aplnrs. Recently, a novel peptide hormone named Elabela (Ela, also known as Apela or Toddler) was identified in zebrafish and shown to bind and activate Aplnrs (*Chng et al., 2013*; *Pauli et al., 2014*). By in situ hybridization we detected *ela* expression by the NC (*Figure 2—figure supplement 1D*, [*Pauli et al., 2014*]), consistent with a possible role in attracting angioblasts to the midline. We next analyzed angioblast migration in zebrafish embryos carrying a homozygous null mutation in the *ela* gene (*ela$^{br13}$*). Ela deficiency led to a strong impairment of angioblast migration (*Figure 2A*), indicating that indeed Ela-Aplnr represent a novel *bona fide* ligand–receptor pair and therefore a novel signaling pathway regulating angioblast migration.

Given that the *ela$^{-/-}$* phenotype was not as severe as the depletion of both receptors and that *apln* expression increases later in the NC (while *ela* expression becomes progressively reduced, *Figure 2—figure supplement 1C,D*), we analyzed the phenotypes of the offspring of *apln$^{+/-}$;ela$^{+/-}$* double heterozygous parents. The embryos were all individually scored for their phenotype regarding angioblast midline migration into four categories (normal, mild, strong, severe) and then subjected to genotyping. Interestingly a mild phenotype was observed, when either one *ela* or both *apln* alleles were mutant, a strong phenotype when *ela* was homozygously mutant with increasing severity (severe phenotype) upon loss of the *apln* alleles (*Figure 2D,E*). Homozygous *ela$^{-/-}$; apln$^{-/-}$* double mutants phenocopied apln receptor deficiency (compare in *Figure 2A,B*). The observed dose dependency is indicative, that in this context Ela and Apln might act as novel chemoattractants for angioblast migration to the midline, with Ela acting as the major endogenous attractant and Apln having a minor additive role.

Previous analysis of zebrafish embryos lacking the NC showed, that signals from the midline guide angioblast migration (*Fouquet et al., 1997*; *Sumoy et al., 1997*). We could show that

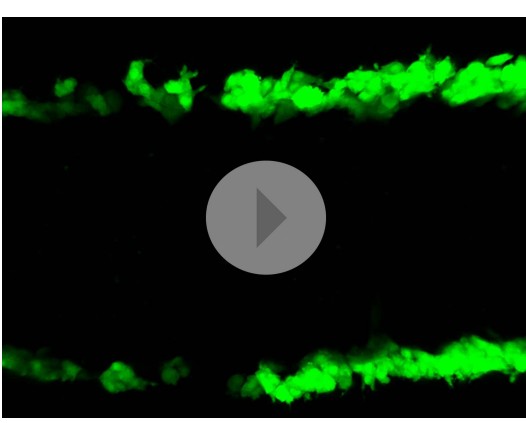

**Video 2.** Angioblast migration between 12.5 hpf and 16 hpf observed by time-lapse imaging using *Tg(fli1a:EGFP)$^{y1}$* to visualize angioblasts. No migration to the midline, but minor movements and strong filopodia formation can be visualized in Aplnra/b double deficient embryos (*aplnra/b* MO).

indeed angioblast migration as well as *ela* expression is perturbed in the zebrafish mutants lacking a NC (*notochord homeobox, noto*; previous name: *floating head*) or exhibiting defective NC precursor differentiation (*T, brachyury homolog a, ta*; previous name: *no tail*) (*Figure 3*) (*Halpern et al., 1993*; *Odenthal et al., 1996*). While angioblasts in *noto^{tk241}* mutant embryos failed to migrate to the midline, *ta^{b160}* angioblasts migrated towards the midline but failed to reach this structure (*Figure 3A*). In line with these results, *ela* expression was strongly diminished in the *ta^{b160}* midline and almost absent in *noto^{tk241}* mutant embryos (*Figure 3B*). We next wanted to test whether midline *ela* expression would be sufficient to restore angioblast migration in *ta^{b160}* mutants. For this purpose, plasmid DNA with a *shh* promoter fragment enabling *ela* expression in the floorplate was injected together with *Tol2* transposase mRNA to obtain mosaic Ela overexpression in the midline. Indeed, Ela overexpression did restore angioblast migration to the midline in *ta^{b160}* embryos (*Figure 3C*), demonstrating that midline Ela expression is not only necessary, but sufficient to regulate angioblast migration during vasculogenesis.

Next, we wanted to analyze the function of Ela-Aplnr signaling more mechanistically. Recently, it was published that Ela does not act as a chemoattractant, but instead acts as a motogen in regulating mesoderm migration during gastrulation (*Pauli et al., 2014*). In contrast, the *aplnr* deficient (*aplra/b* MO injected) angioblasts were still partially motile and did send out filopodia, but failed to migrate to the midline (*Video 2*). To test whether Ela does act as a chemoattractant or just enables motility of angioblasts, we overexpressed Ela at ectopic locations in *noto* or *ta* mutant embryos, as both of these lack endogenous *ela* and *apln* expression in the NC. We used a heatshock promoter to control timing of expression and labeled Ela overexpressing cells by coexpression of RFP through an internal ribosome entry site (IRES) element (*Figure 4*). We measured the distance of control (Cherry) or Ela/RFP expressing cells from angioblasts in a 40 µm radius of the red labeled cell and considered an

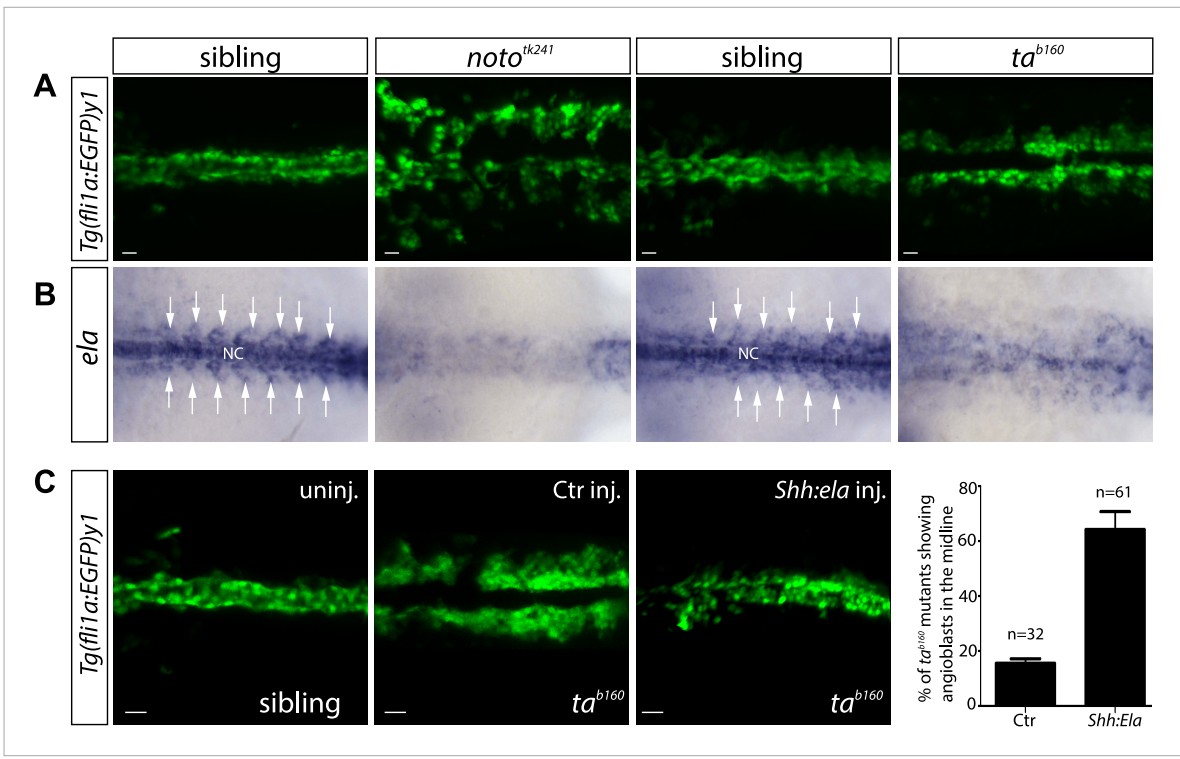

**Figure 3**. Notochord (NC) *ela* expression is sufficient to guide angioblasts to the midline. (**A**) Angioblasts fail to migrate to the midline position in the NC mutants *noto* and *ta*. Confocal projections of *Tg(fli1a:EGFP)^{y1}* embryos in dorsal view at 17 hpf (scale bars: 20 µm). (**B**) *In situ* hybridization in 17 hpf old zebrafish embryos showing *ela* expression from the NC and somites (white arrows). NC deficiency abolishes or strongly reduces *ela* expression in *noto* or *ta* mutant embryos. (**C**) Mosaic expression of Ela in the midline, achieved by injection of *shh:ela* DNA, led to rescue of angioblast migration in *ta^{b160}* mutant embryos (n = 61 embryos, ctr injected n = 32 embryos). Significance was calculated by chi-square ($\chi^2$ = 19.65) equivalent to p < 0.001 (p = 9E-06). Scale bars: 30 µm.

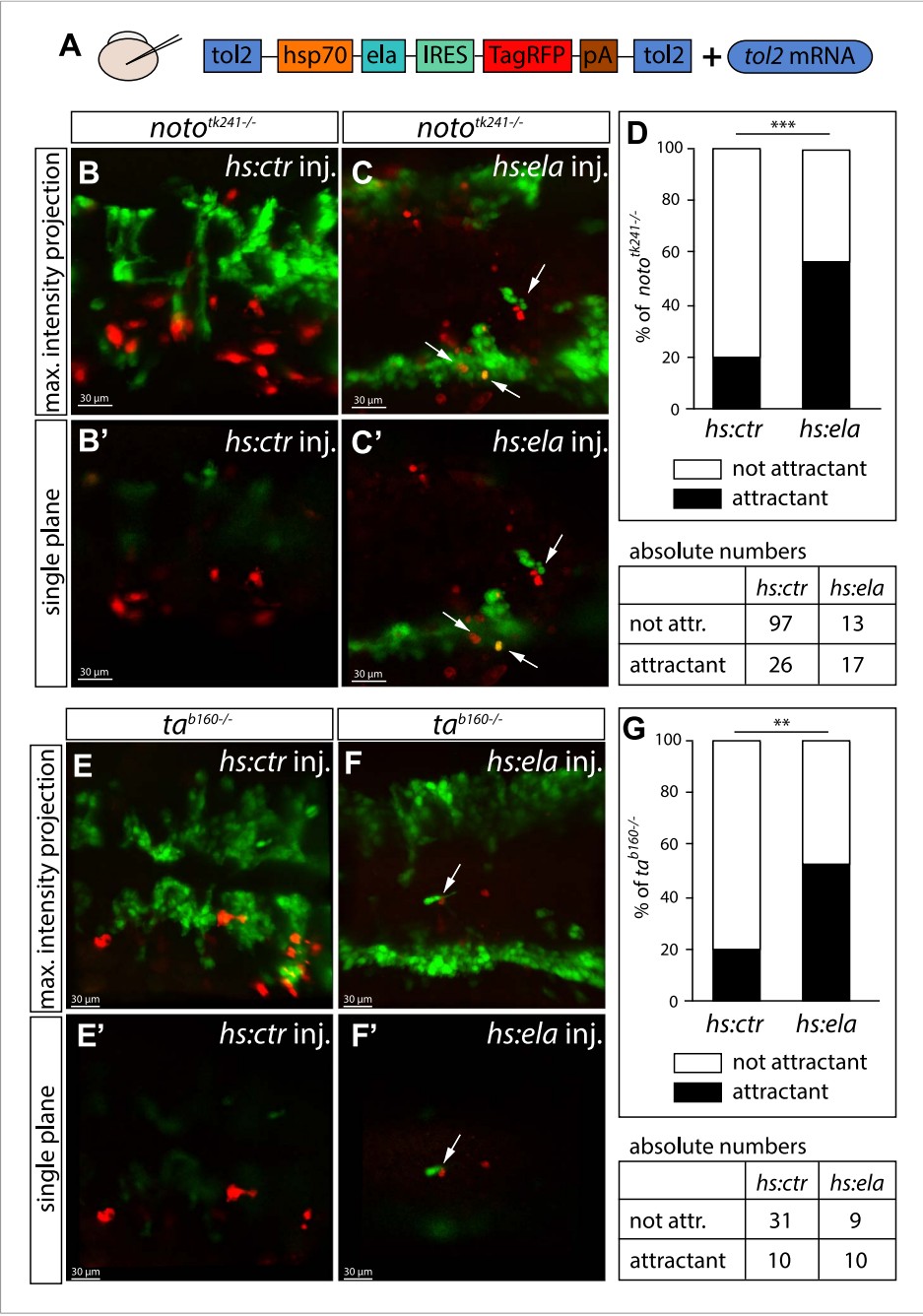

**Figure 4**. Ela overexpressing cells attract angioblasts. (**A**) F1 embryos from *noto*[tk241+/−] or *ta*[b160+/−] parents were injected with 250 pg *Tol2* mRNA as well as 10 pg of DNA constructs, in which the heatshock promotor was used to drive either *control (ctr, Cherry)* or *ela* expression. Individual Ela overexpressing cells were labeled by IRES mediated RFP expression. Expression was induced by two consecutive heatshocks (incubation at 39°C) at 12 hpf and 14 hpf for 1 hr each. (**B–C'**) Angioblast migration in *noto*[tk241−/−] mutant embryos injected with the *ctr* (**B, B'**) or the *ela* overexpression (**C, C'**) construct. (**D**) Quantification of Ela/RFP or ctr (Cherry) positive cells showed significantly more Ela overexpressing cells attracting angioblasts than ctr cells (n = 10 embryos for *hs:ela*, n = 20 embryos for *hs:ctr* ; p*** = 0.0001). (**E–F'**) Angioblast migration in *ta*[b160 −/−] mutant embryos injected with the ctr (**E, E'**) or the *ela* overexpression (**F, F'**) construct (**G**) Quantification of Ela/RFP or ctr (Cherry) positive cells showed significantly more Ela overexpressing cells attracting angioblasts than ctr cells (n = 8 embryos for *hs:ela*, n = 10 embryos for *hs:ctr* ; p** = 0.0089). Significance was calculated by chi-square test. Maximum intensity projections give an overview of the analyzed embryos. Single planes visualize the closeness of Ela or ctr (Cherry) expressing cells to angioblasts. White arrows point to Ela overexpressing cells with less than 5 μm distance to angioblasts. <5 μm distance between an

*Figure 4. continued on next page*

*Figure 4. Continued*

Ela/ctr positive cell and an angioblast was counted as 'attractant'; 5–40 µm distance was counted as 'not attractant'. Scale bars represent 30 µm.

angioblast as attracted, if it was less than 5 µm away from it. While only 20% of the angioblast were found in the vicinity of control cells, more than 52% of the angioblasts were close to Ela overexpressing cells (*Figure 4D,G*). Together with the observed dose dependency of Ela-Apln, the motility of Aplnr deficient embryos and the rescue of NC mutants by midline Ela expression, our data support the interpretation, that Ela acts as a chemoattractant for angioblasts during midline migration.

To prove that Ela-Aplnr signaling is directly required in angioblasts and rule out that the observed migration defects were caused by changes in other cell types, we performed transplantation experiments. Mini-ruby injected (red fluorescence) transgenic *fli1a:EGFP* positive (angioblasts, green) cells from control MO injected, *aplnra/b* MO injected or wild type (WT) donor embryos were transplanted into either WT or *aplnr* loss-of-function (*apnlra/b* MO injected) host embryos (*Figure 5A*). Analysis of angioblast migration to the midline at 17 hpf revealed that in average 93.6% of control MO-injected transplanted angioblasts migrated to the midline in a WT host embryo (*Figure 5B,C*; n = 39 GFP+ angioblasts, 7 embryos). In contrast, only 9.4% of *aplnra/b* MO injected angioblasts arrived at the midline, whereas the majority of these cells remained in more lateral positions (*Figure 5B,D*; n = 62 GFP+ angioblasts, 5 embryos). Transplanted WT angioblasts were perfectly capable to migrate to the midline in an *aplnr* deficient environment (average of 81.9%, *Figure 5B,E*; n = 15 GFP+ angioblasts, 6 embryos), which shows that there is indeed a cell autonomous requirement for Aplnr signaling in angioblasts.

Based on the sum of the results presented above, we propose that Ela expression at the embryonic midline provides a chemoattractive signal for Aplnr-positive angioblasts. While Ela and Apln both signal to the Apln receptors, the initial *ela* expression at the midline appears stronger than *apln*, which peaked at later developmental time points (*Figure 2—figure supplement 1C,D*). Thus, *ela* and *apln* may have partially redundant functions, but are required at different stages of vascular morphogenesis. Surprisingly, while Ela-Aplnr signaling was strictly required for the guided migration of angioblasts, this process was not controlled by Vegfa-mediated signaling.

Our studies identified Ela-Aplnr signaling as a novel signaling pathway for angioblasts regulating vascular patterning in the developing vertebrate embryo. As therapeutic intervention with vascular growth is clinically highly relevant and, so far, predominantly focuses on the VEGF-A signaling cascade, the Ela-Aplnr signaling pathway may also represent a novel therapeutic target in human disease.

# Materials and methods

## Zebrafish husbandry and strains

Zebrafish (*Danio rerio*) were maintained in a recirculating aquaculture system under standard laboratory conditions (*Westerfield, 1993*). Embryos were staged by hours post fertilization (hpf) at 28.5°C (*Kimmel et al., 1995*), for 17 hpf embryos were incubated after gastrulation at 21°C and staged on the following morning by counting somites (16 somites equal 17 hpf [*Kimmel et al., 1995*]).

Zebrafish strains used were Tg(fli1a:EGFP)[y1] (*Lawson and Weinstein, 2002*), kdrl[hu5088] (*Bussmann et al., 2010*), ta[b160] (*Halpern et al., 1993*), noto[tk241] (*Odenthal et al., 1996*) and ela[br13] (*Chng et al., 2013*).

## Generation of *apln*[mu267], *aplnra*[mu296] and *aplnrb*[mu270] mutants

For *apln* and *aplnrb* Crispr mediated mutagenesis was used. Nls-zCas9-nls mRNA was synthesized as previously described (*Jao et al., 2013*).

The target sequences of *apln* exon 1 (5′-GAATGTGAAGATCTTGACGC-3′) and *aplnrb* exon 1 (5′-CTACATGCTCATCTTCATCC-3′) were cloned into the gRNA expression vector pDR274 as previously described (*Hwang et al., 2013*). The *apln* sgRNA or *aplnrb* sgRNA were transcribed using DraI-digested gRNA expression vector as template and the T7 mMessage mMachine kit (Ambion, Life Technologies, Germany). *apln* sgRNA or *apnrb* sgRNA and nls-zCas9-nls-encoding mRNA were co-injected into one-cell stage zebrafish embryos. Each embryo was injected with 2 nl of solution containing 12.5 ng/µl sgRNA and 300 ng/µl nls-zCas9-nls mRNA. A HgaI (New England BioLabs, Frankfurt, Germany) restriction site was used for genotyping of *apln*[mu267] and a FokI (New England BioLabs) restriction site was used for genotyping of *aplnrb*[mu270].

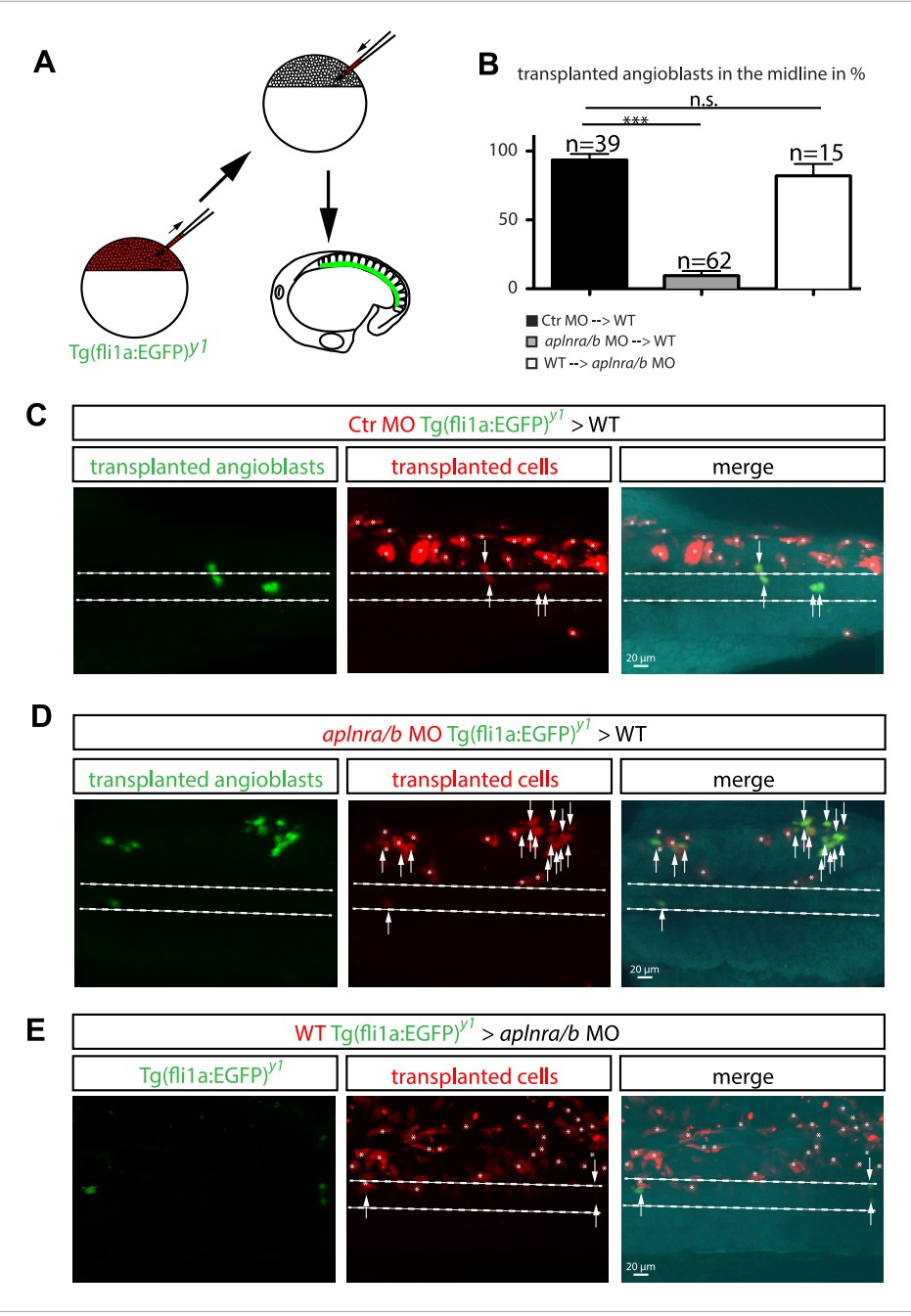

**Figure 5**. Cell autonomous requirement for Apelin receptor signaling in angioblasts. (**A**) Experimental design: mini-ruby injected cells from ctr MO (**C**) *aplnra/b* MO (**D**) or WT (**E**) *Tg(fli1a:EGFP)*^y1^ embryos were transplanted into WT or *aplnra/b* MO host embryos and scored for their migration to the midline. (**B**) Quantification: 93.6% of *ctr* MO injected (n = 39 GFP + angioblasts, 7 embryos), but only 9.4% of *aplnra/b* deficient (*aplnra/b* MO injected; n = 62 GFP+ angioblasts, 5 embryos) donor angioblasts migrated to the midline in WT host embryos. In contrast, 81.9% of WT donor angioblasts (n = 15 GFP+ angioblasts, 6 embryos) migrated to the midline in *aplnra/b* deficient host embryos. Error bars represent SEM, calculated using the standard deviation of percent of angioblasts in the midline per embryo. Statistical analysis showed significance using 2 way ANOVA and t-test, with p *** = 3.44577E-08 for *aplnra/b* MO in WT, and p = 0.232995135 (not significant, n.s.) for WT in *aplnra/b* MO. see also *Figure 5—source data 1*. (**C**–**E**) Confocal projections showing representative embryos of the transplantation experiments at 17 hpf. Arrows indicate transplanted angioblasts; asterisks label transplanted cells, which are not angioblasts; dashed lines indicate the midline.

*Figure 5 continued on next page*

*Figure 5 Continued*

The following source data is available for figure 5:

**Source data 1**. Statistical analysis and single values for the number of angioblasts and embryos analyzed.

---

aplnra mutants were generated by TALEN mutagenesis. TALENs were assembled using the Golden Gate method (*Cermak et al., 2011*). For targeting the *aplnra* locus, a 5′ RVD (NI HD NI HD HD NH NI NH NI HD NI NG NI HD NH NI NG) and a 3′ RVD (HD NI HD NI HD HD HD NI NH NI NH NG HD NI NG NG NI NG NI) were generated. A BsrI (New England BioLabs) restriction site was used for genotyping of *aplnra^{mu296}*. mRNA was generated using the T3 mMessage mMachine Kit (Ambion) and injected using 100 pg of the TALEN mix.

## Microinjections

Microinjections of mRNA or MOs were performed as previously described (*Nasevicius and Ekker, 2000*).

mRNA was transcribed using SP6 polymerase (Sp6 mMessage mMachine kit [Ambion]). 100 pg *H2B-cherry* mRNA (*Santoro et al., 2007*), 2 ng *aplnra* MO (*Scott et al., 2007*), 0.5 ng *aplnrb* MO (*Zeng et al., 2007*), 2 ng *apln* MO (*Scott et al., 2007*) or 2 ng *vegfaa*/ 2.3 ng *vegfab* MO (*Ober et al., 2004*) were injected. Simultaneous knockdown of *aplnra* and *aplnrb* was done by coinjection of 2 ng *aplnra* MO and 0.5 ng *aplnrb* MO (referred to as *aplnra/b* MO). 4 ng of standard control MO (Gene tools, Philomath, Oregon) were injected as control.

Plasmid DNA was injected together with *Tol2* mRNA (*Kawakami et al., 2004*). A 2.2 kb fragment of the *shh* promoter (*Gordon et al., 2013*) was used to drive either *rfp* or *ela* expression (*shh:ctr* or *shh:ela*). The *heatshock* promoter (*hsp70l*) was used to drive either *Cherry* (*hs:ctr* [*Hesselson et al., 2009*]) or *ela-IRES-RFP* (*hs:ela*).

## Whole mount in situ hybridization

In situ hybridizations were performed as previously described (*Helker et al., 2013*) using the following probes: *cdh5* (*Larson et al., 2004*), *etv2* (*Helker et al., 2013*), *vegfc* (*Hogan et al., 2009*), *vegfaa* (*Lawson et al., 2002*).

*apln, aplnra* and *aplnrb* probe templates were amplified from cDNA of 19 somite stage old embryos using the following primers: *apln*-forward 5′-GAAAGGCCCAAGTCACAGAG-3′ and *apln*-reverse 5′-GAGTTCACTATCTGATGTCAAACCA-3′, *aplnra*-forward 5′-GAAAGGCCCAAGTCACAGAG-3′ and *aplnra*-reverse 5′-GAGTTCACTATCTGATGTCAAACCA-3′, *aplnrb*-forward 5′-GAAAGGCCCAAGTCACAGAG-3′ and *aplnrb*-reverse 5′-GAGTTCACTATCTGATGTCAAACCA-3′.

The T7 promotor was added to the 5′- end of the reverse primer in a second round of amplification (T7-*apln*-reverse 5′-GTAATACGACTCACTATAGGGAGTTCACTATCTGATGTCAAACCA-3′, (T7-*aplnra*-reverse 5′-GTAATACGACTCACTATAGGGAGTTCACTATCTGATGTCAAACCA-3′, (T7-*aplnrb*-reverse 5′-GTAATACGACTCACTATAGGGAGTTCACTATCTGATGTCAAACCA-3′).

## In vivo time-lapse analysis and confocal microscopy

Embryos were manually dechorionated and mounted in 0.3% agarose (with subsequent removal of agarose from the head/tail region). Medium and agarose were supplemented with 19.2 mg/l Tricaine and 30 mg/l Phenylthiourea. For time-lapse imaging, embryos were kept in a 28.5°C heated chamber surrounding the microscope stage. All fluorescent images were acquired using an upright Leica Sp5 DM 6000 or a Zeiss LSM 780 Confocal microscope. Confocal stacks and confocal movies were assembled using Imaris Software (Bitplane, Switzerland).

## Transplantation experiments

Donor embryos were injected with *ctr* MO or *aplnra/b* MO and labeled by the injection of 0.1 ng mini-Ruby (Tetramethylrhodamine, Invitrogen/Life Technologies). At sphere stage, cells were removed from donor embryos and transferred to wild type hosts using a glass capillary. Transplanted angioblasts were identified by transgenic EGFP expression together with mini-Ruby stain. By counting the number of EGFP positive donor angioblasts in the midline vs all EGFP positive donor angioblasts (100%), the percentage of donor angioblasts in the midline was determined for each individual embryo.

## Acknowledgements

We thank Aurelien Courtois and Jeroen Bussman for helpful discussion and suggestions. We thank Arnd Siekmann for the *vegfaa* and *vegfab* MO and Stefan Schulte-Merker for providing plasmid containing the *shh* promoter and the IRES element. We thank Ralf Adams for critical comments on the manuscript. We thank Stefan Volkery for imaging advice and microscope maintenance. Excellent fish husbandry and technical assistance were provided by Reinhild Bußmann and Katja Müller.

## Additional information

### Funding

| Funder | Grant reference | Author |
|---|---|---|
| State of Northrhine Westphalia | Northrhine Westphalia Return Fellowship | Wiebke Herzog |
| Westfälische Wilhems-Universität Münster | Cells in Motion cluster of Excellence, CiM - EXC 1003 / FF-2013-14 | Wiebke Herzog |
| Agency for Science, Technology and Research (A*STAR) | Strategic Positioning Fund on Genetic Orphan Diseases | Bruno Reversade |
| Max-Planck-Gesellschaft | | Friedemann Kiefer |

The funders had no role in study design, data collection and interpretation, or the decision to submit the work for publication.

### Author contributions

CSMH, Conception and design, Acquisition of data, Analysis and interpretation of data, Drafting or revising the article; AS, Acquisition of data, Analysis and interpretation of data, Drafting or revising the article; CP, FK, Analysis and interpretation of data, Drafting or revising the article; SCC, Contributed preliminary data, Drafting or revising the article, Contributed unpublished essential data or reagents; BR, Drafting or revising the article, Contributed unpublished essential data or reagents; WH, Conception and design, Analysis and interpretation of data, Drafting or revising the article

### Ethics

Animal experimentation: All animal experiments were performed in strict accordance with the relevant laws and institutional guidelines the Max Planck Institute for Molecular Biomedicine, Muenster and the Institute of Medical Biology, Singapore. All protocols were approved by animal ethics committees of the state of North Rhine-Westfalia (Germany,# 39.32.7.1) and Singapore, respectively, and all efforts were made to minimize suffering.

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
