## [Decision Letter]

Thank you for sending your work entitled "The hormonal peptides Elabela and Apelin guide angioblasts to the midline during vasculogenesis" for consideration at *eLife*. Your article has been evaluated by Janet Rossant (Senior editor), Tanya Whitfield (Reviewing Editor), and two additional reviewers.

The Reviewing editor and the other reviewers discussed their comments before we reached this decision, and the Reviewing editor has assembled the following comments to help you prepare a revised submission.

While the reviewers have all found the work interesting, all three have concerns over the extent that the data support your conclusions, including the challenges that you propose to existing literature. The full reviews are appended below, but a summary of the main points that must be addressed is:

1) Data should be supplied to demonstrate the efficacy of each of the morpholinos used, especially (in the case of the Vegf study) when the authors are proposing that knockdown has no effect on angioblast migration. The authors should also show whether any of the morpholinos cause systemic non-specific effects, such as developmental delay. If mutants are available, it would be helpful to repeat the analysis in mutants.

2) Additional, and quantitative, data are needed to support the assertion that Ela acts as an instructive chemoattractant in this context and not simply as a motogen. As two of the reviewers have pointed out, the dataset is currently too small to be able to reach this conclusion. Further data should be supplied to illustrate this point.

It is vitally important to provide strong support for the two points above, as these are the proposed advances in the understanding of angioblast migration over previously published literature.

3) The relative contributions of Apln and Ela to angioblast migration should be clarified, both in the Results section and in the Abstract and Impact Statement.

4) A reduction in Shh signalling should be demonstrated to support the cyclopamine data.

5) Take care to acknowledge the findings of previously published work e.g. [26], and previously published expression patterns, appropriately.

6) Improve quantitation of all experiments: provide cell counts to support image panels; state n numbers wherever % values are shown, and ensure that appropriate statistical tests have been applied.

7) Correct typos, formatting and gene nomenclature throughout.

Reviewer #1:

This is an interesting and well-written short report examining the role of the Apelin receptor signalling pathway in angioblast migration in the zebrafish embryo.

The study challenges the current VEGF model for angioblast migration to the midline, and also proposes that in this context, the ligand Elabela functions as a chemoattractant rather than a motogen, again challenging previous interpretations in the published literature. Although the authors present some interesting preliminary observations to support these challenges, the data need further analysis and quantitation to strengthen the authors' conclusions.

The paper relies on morpholino (MO) knockdown for a number of experiments. Given the current scepticism in the field surrounding the use of MOs, it is particularly important to demonstrate that the MO experiments are justified and rigorous. The authors give references to previously published work for each MO they use, but nevertheless it would be helpful to show data (e.g. antibody or RNA splicing data demonstrating specific knockdown), or give a statement, to comment on the specificity and efficacy of each MO.

Confidence in the MO data is especially important given that the main phenotype that is shown for many of the experiments (a failure of angioblasts to migrate to the midline) could also be explained by a generalised developmental delay. It is therefore important to demonstrate that the MOs do not cause any general developmental delay (e.g. by showing a brightfield image of the whole embryo, and providing quantitative information about, for example, somite number).

The challenge to the VEGF model, a major conclusion of the manuscript, is based on just two data panels (Figure 1), with no accompanying quantitation. For the *vegfaa* MO experiment, notwithstanding the caveats mentioned above for MO use, there do appear to be a few green cells beyond the midline, but the picture is so heavily cropped that it is not possible to assess this properly. The authors should provide some quantitation, together with further information to show that the MO is indeed disrupting VEGF signalling, to strengthen their conclusions for this experiment. The authors state that *vegfaa* is the 'main ortholog' in zebrafish, but it would also be helpful to know whether there are any other *vegf* genes or isoforms expressed at the relevant time and place, and whether these are affected by *vegfaa* MO knockdown or not. The mutant VEGFR (*kdrl*) analysis is more convincing, but again, are there other genes or isoforms that might function redundantly with the *kdrl* receptor?

The second challenge to existing literature, that Elabela (in this context) acts a chemoattractant rather than a motogen, is also supported only weakly, being based on a single supplementary figure panel (Figure 3–figure supplement 1D) showing one (or possibly two) green angioblasts in an ectopic dorsal position near to the ectopic Elabela-expressing (red) cells. This needs some further examples and quantitiation to support the authors' conclusion.

Quantitation:

The % data shown in Figures 2, 3 and 4 must be accompanied by corresponding n numbers, preferably shown directly on the graphs.

Figure 4. A *t*-test would not be recommended for an experiment involving more than one experimental condition. This analysis should be repeated using ANOVA or an equivalent test. The graph should use a bar to show which columns are being compared (presumably each of the experimental situations with the control).

Reviewer #2:

Helker C. et al. demonstrate the essential role for Elabela (Ela)-Apelin (Apln) receptor (Aplnr) signaling in the initial assembly of angioblasts as a vascular cord in the midline. Unexpectedly, *vegfa* is not required as a chemoattractant for angioblast migration, while Elabela, a secretary peptide from the notochord, functions as an angioblast chemoattractant. The data has been convincingly demonstrated using several gene mutants (*krdl*^-/-^, *ela*^br13-/-^, *apln*
^mu267-/-^, and *noto*) and their morphant.

The objectives written and the findings shown in the present manuscript are clear; however, there are several points should be addressed. There are several statements that are not supported by their results in the present study.

Major concerns:

1) Requirement of Apln for angioblast migration:

The authors claim that not only Ela but also Apln guides angioblasts to the midline in the Results section and in the Title. In Figure 2, the data of *apln*^*mu267-/-*^ clearly show that Apln is not required for angioblast migration. However, the authors point to the requirement of Apln using the embryos of the *ela*^*-/-*^ and *apln*^*-/-*^. On the contrary, they state the impact of Ela (instead of Apln)-Aplnr signaling in the Impact Statement. This interpretation might puzzle the readers. Therefore, they need to re-interpret the data and re-write the manuscript including the Title.

2) Vegf or Vegfa:

The authors use "Vegf" and "Vegfa" in the Abstract and main text. This inconsistency should be corrected not to mislead the readers. Are other *vegf* members except Vegfa are thought to regulate angioblast migration in the previous papers? According to their results, the conclusion might be that Vegfa-Vegfr2 signaling is required for angioblast migration toward the midline. Therefore, they should rewrite the manuscript by keeping the consistency when they use the similar words.

Reviewer #3:

Helker and colleagues provide evidence for new roles of previously identified small peptides Apelin and Elabela in guiding angioblasts towards the embryonic midline during zebrafish embryogenesis. Previous studies implicated Vascular endothelial growth factor in convergence of angioblasts towards the embryonic midline, where they coalesce to build the first embryonic vessels. The authors first present evidence from zebrafish against the requirement for Vegf in angioblast migration towards the midline. Rather, they proposed, Apelin and Elabela, have partially redundant function to guide angioblasts to the midline by acting through the Apelin receptors. Apelin, Apelin GPCR Receptor axis has been previously implicated in cardiac progenitor cell convergence and cell fate specification, by Scott et al., and Zeng et al., in 2007. Elabela acting through Apelin receptors has been shown to be required for gastrulation cell movements and both heart and endoderm development by [4] and [26]. However, neither Apelin nor Elabela have been proposed to affect gastrulation cell movements acting as guidance molecules, with Elabela being considered a motogen. The authors employ morpholino oligonucleotides and mutations to show that Apelin and Elabela expressed in the notochord have partially redundant function in angioblast migration. Moreover, *apelin* receptors *a* and *b* expressed in angioblasts also have redundant function in this process. Therefore the proposed role of Apelin/Elabela/Apelin receptor signaling to guide angioblasts towards the embryonic midline would be a significant advance. However, the evidence in support of these conclusions are not compelling. There are also problems with the results and their interpretations. These problems should be addressed before the manuscript becomes suitable for publication.

The most novel conclusion of this work is that Elabela/Apelin/Apelin receptor axis has an instructive role in angioblast migration, by guiding them to the midline. The key line of evidence is that when cells ectopically expressing Ela were present in Ela mutant: "We found that individual Ela overexpressing cell were indeed capable of attracting angioblast migration to ectopic locations […]. our data therefore demonstrate, that Ela acts as a chemoattractant for angioblasts during midline migration".

The presented data is not compelling. The authors observed occasional overlap between angioblasts and these Ela expressing cells. An alternative interpretation is that Ela expressed in ectopic location increased motility of angioblasts and some of them wondered in ectopic locations.

The conclusion about Vegf signaling not being required in angioblast migration is based on the observation that "neither Vegf ligand depletion (*vegfaa* MO) nor a null mutation in the gene encoding the VEGFR2 ortholog (*kdrl*) interfered with angioblast migration in zebrafish embryos." However, there is no data showing the degree of loss-of-function for morpholino or for *kdrl* mutant.

Likewise, when the authors inhibit *Shh* signaling using cyclopamine, no quantitative data is presented documenting the degree of *Shh* signaling inhibition. Reduced expression of *vegf* is shown by in situ hybridization, but this is not a quantitative results. Such body of evidence is not sufficient to support the authors' conclusion regarding:

Does mutation in *apelin* gene affect *elabela* expression and vice versa?

Do mutations in *kdrl* affect phenotypes of *apelin*, *elabela* or *apelin* receptor single mutants?

The authors conclude that simultaneous knockdown of *aplnra* and *b* affects angioblast cell movement but not specification, based on morpholino interference. Whereas previously published morpholinos are used, appropriate experiments showing efficiency and specificity should be presented. Given that mutations for these receptors are available, they should be preferably employed.

Similar concerns are regarding *apelin* morpholino and *apelin* mutant experiments. Does *apelin* mutation lead to nonsense mediated degradation of *apelin* mRNA?

The authors conclude that angioblast migrate normally in *apelin* mutants, however, in Figure 2 it appears that ca. 30% of *apelin* mutants showed mild angioblast migration phenotype. This along with a strong enhancement of the phenotype of *elabela* mutant indicates that *apelin* is also required for angioblast migration.

In the Results the authors write that cardiomyocyte deficiency observed in *aplnr* deficient embryos could not be phenocopied by loss of *apelin* function. This is not correct; Scott et al., and Zeng et al. reported reduction of cardiomyocyte formation in *apelin* morphant embryos.

[Editors' note: further revisions were requested prior to acceptance, as described below.]

Thank you for resubmitting your work entitled "The hormonal peptide Elabela guides angioblasts to the midline during vasculogenesis" for further consideration at *eLife*. Your revised article has been favorably evaluated by Janet Rossant (Senior editor) and Tanya Whitfield (Reviewing Editor). The manuscript is much improved and most of the reviewers' recommendations have been addressed. Mutant phenotypes have been used to back up the morpholino data, and most of the requested control data have been provided to demonstrate the efficacy of morpholinos and compounds used. Description of relative roles for Apln and Ela in the migration of angioblasts is now much clearer. However, there are still some minor issues concerning the description of the experimental design and quantitation of the data that should be addressed before publication, as outlined below:

1) The lack of a generalised developmental delay (control for somite number) should be stated explicitly in the text as well as in the Materials and methods, so that it can be ruled out as a possible explanation for a delay or failure of angioblasts to migrate to the midline in MO-injected or mutant embryos.

2) The new data supporting the assertion that Ela acts as a chemoattractant and not a motogen are still based on rather low numbers, and indeed the new data are still presented only as a supplementary figure (Figure 3–figure supplement 1). In the text description (Results, seventh paragraph), n numbers (cells and embryos) must be quoted in the text to support the percentage values, in addition to the information provided in the figure.

The chemoattractive effect of Ela appears to be an important conclusion of the manuscript, but is only based on this supplementary figure. If the data are not strong enough to be shown as a main figure, the conclusion should also be toned down: rather than stating that the data 'demonstrate' that Ela acts as a chemoattractant, wording such as 'the data support the interpretation that Ela is acting as a chemoattractant in this context, but further experiments and higher n numbers would be required to test this more rigorously' should be used. In addition, use of the word 'chemoattractant' should be treated with similar caution throughout the manuscript.

3) Figure 4 and description and Results. Again, n numbers should be stated along with the percentages here, in the text as well as on the figure.

4) What do the error bars represent in Figure 4?

---

## [Author Response]

*1) Data should be supplied to demonstrate the efficacy of each of the morpholinos used, especially (in the case of the Vegf study) when the authors are proposing that knockdown has no effect on angioblast migration. The authors should also show whether any of the morpholinos cause systemic non-specific effects, such as developmental delay. If mutants are available, it would be helpful to repeat the analysis in mutants*.

We have added mutant analysis, to complement morpholino data, wherever possible.

We generated new mutants for both Apln receptors as well as double mutants and have provided better controls as well as additional pharmacological studies for the analysis of the role of Vegf signaling.

Figure 1 has been changed and is now showing *vegfaa/vegfab* double MO injections, so that they cannot compensate for each other.

In addition a new supplemental figure, Figure 1—figure supplement 1, has been generated (the cyclopamine data are now Figure 1—figure supplement 2). We are now showing the later phenotypes of Vegf-signaling deficiency, to demonstrate efficiency of the used reagents. Also inhibition of Vegf-signaling by chemical treatment has been added as an alternative to MO generated data.

Figure 2: new panels have been added and show the phenotype-genotype correlation of single and double *apelin* receptor mutant embryos.

Figure 2—figure supplement 2: generation of the *aplnr* mutants has been added.

Any potential effect of developmental delay has been avoided by careful staging of the embryos (i.e. counting of somites to ensure the same developmental stage: now included in the method description).

*2) Additional, and quantitative, data are needed to support the assertion that Ela acts as an instructive chemoattractant in this context and not simply as a motogen. As two of the reviewers have pointed out, the dataset is currently too small to be able to reach this conclusion. Further data should be supplied to illustrate this point*.

We have added a new series of experiments, ectopically overexpressing Ela or Cherry in *noto* or *ta* mutants (to reduce endogenous Ela and Apln in the midline), in which the distance of angioblasts to Ela overexpressing cells (compared to Cherry overexpressing cells) was quantified.

Our data strongly support the role of Ela as a chemoattractant in the context of angioblast migration.

To address this point, a new Figure 3–figure supplement 1 has been added.

*It is vitally important to provide strong support for the two points above, as these are the proposed advances in the understanding of angioblast migration over previously published literature*.

*3) The relative contributions of Apln and Ela to angioblast migration should be clarified, both in the Results section and in the Abstract and Impact Statement*.

We have changed the manuscript text accordingly.

*4) A reduction in Shh signalling should be demonstrated to support the cyclopamine data*.

Block of hedgehog signaling using cyclopamine was originally published in chicken embryos (Incardona JP et al. Development. 1998 Sep;125(18):3553-62).

In zebrafish cyclopamine treatment has been used in many more than 70 publications, most notably it was shown that:

a) 200µM treatment fully phenocopied the *shh* mutant *sonic you* (Neumann CJ et al, Development. 1999 Nov;126(21):4817-26).

b) 100µM completely eliminated Hh activity (Chen W et al., Development. 2001 Jun;128(12):2385-96.

c) 25µM cyclopamine completely abolishes Vegf expression in the somites (as also seen in *syu* and *yot* mutants) (Lawson ND et al Dev Cell. 2002 Jul;3(1):127-36).

As this analysis of cyclopamine effects has therefore been done extensively and we have used 100µM (far more than N. Lawson, who demonstrated the effect on Vegf expression), we have not added in situs on other downstream markers. We have changed the wording, to state the previously published result more clearly.

However, we have added an alternative experiment showing the requirement for Vegf signaling by treatment with a Vegf receptor 2 inhibitor, hoping that this will be provide a meaningful alternative (see Figure 1—figure supplement 1).

*5) Take care to acknowledge the findings of previously published work e.g.*
[26]*, and previously published expression patterns, appropriately*.

We have revised the manuscript and added the appropriate references accordingly,

*6) Improve quantitation of all experiments: provide cell counts to support image panels; state n numbers wherever % values are shown, and ensure that appropriate statistical tests have been applied*.

We have added the n-numbers (cells and embryos) for all quantifications, provided source files, and added statistical analysis as requested.

*7) Correct typos, formatting and gene nomenclature throughout*.

We have edited the manuscript carefully, with special attention to these issues. It sometimes remains non-intuitive for the reader, that the nomenclature rules are not the same for mouse and zebrafish, so we have chosen zebrafish nomenclature, unless specifically referring to mouse data. As an example: "VEGF" would be the correct mouse specific spelling (see http://www.informatics.jax.org/mgihome/nomen/gene.shtml), "Vegf" would be correct for zebrafish (see ZFIN at https://wiki.zfin.org/display/general/ZFIN+Zebrafish+Nomenclature+Guidelines).

*Reviewer #1*:

*The study challenges the current VEGF model for angioblast migration to the midline, and also proposes that in this context, the ligand Elabela functions as a chemoattractant rather than a motogen, again challenging previous interpretations in the published literature. Although the authors present some interesting preliminary observations to support these challenges, the data need further analysis and quantitation to strengthen the authors' conclusions*.

*The paper relies on morpholino (MO) knockdown for a number of experiments. Given the current scepticism in the field surrounding the use of MOs, it is particularly important to demonstrate that the MO experiments are justified and rigorous. The authors give references to previously published work for each MO they use, but nevertheless it would be helpful to show data (e.g. antibody or RNA splicing data demonstrating specific knockdown), or give a statement, to comment on the specificity and efficacy of each MO*.

As discussed above, we have now provided mutant data to complement MO data, wherever possible, for the Vegf studies we have used chemical inhibition of Vegfr2 as an alternative.

*Confidence in the MO data is especially important given that the main phenotype that is shown for many of the experiments (a failure of angioblasts to migrate to the midline) could also be explained by a generalised developmental delay. It is therefore important to demonstrate that the MOs do not cause any general developmental delay (e.g. by showing a brightfield image of the whole embryo, and providing quantitative information about, for example, somite number)*.

We had already staged all embryos according to somite numbers, but failed to state this in the manuscript, we have changed the Methods section accordingly.

*The challenge to the VEGF model, a major conclusion of the manuscript, is based on just two data panels (*Figure 1*), with no accompanying quantitation. For the* vegfaa *MO experiment, notwithstanding the caveats mentioned above for MO use, there do appear to be a few green cells beyond the midline, but the picture is so heavily cropped that it is not possible to assess this properly. The authors should provide some quantitation, together with further information to show that the MO is indeed disrupting VEGF signalling, to strengthen their conclusions for this experiment. The authors state that* vegfaa *is the 'main ortholog' in zebrafish, but it would also be helpful to know whether there are any other* vegf *genes or isoforms expressed at the relevant time and place, and whether these are affected by* vegfaa *MO knockdown or not. The mutant VEGFR (*kdrl*) analysis is more convincing, but again, are there other genes or isoforms that might function redundantly with the* kdrl *receptor*?

*vegfaa* MO experiments have been replaced by combined knockdown of *vegfaa* and *vegfab*. In addition we show as a supplemental figure an analysis of all Vegf-signaling deficient embryos at a later developmental time point, where they all display the previously described defects in Vegf-dependent sprouting angiogenesis.

*The second challenge to existing literature, that Elabela (in this context) acts a chemoattractant rather than a motogen, is also supported only weakly, being based on a single supplementary figure panel (Figure 3–figure supplement 1D) showing one (or possibly two) green angioblasts in an ectopic dorsal position near to the ectopic Elabela-expressing (red) cells. This needs some further examples and quantitiation to support the authors' conclusion*.

As described above a new set of experiments with full quantification has been added as Figure 3–figure supplement 1.

Quantitation:

*The % data shown in*
Figures 2, 3 and 4
*must be accompanied by corresponding n numbers, preferably shown directly on the graphs*.

We have added the n numbers in each graph.

Figure 4*. A* t*-test would not be recommended for an experiment involving more than one experimental condition. This analysis should be repeated using ANOVA or an equivalent test. The graph should use a bar to show which columns are being compared (presumably each of the experimental situations with the control)*.

We have done an ANOVA analysis as well as the *t*-test and added the bars.

*Reviewer #2*:

*Major concerns*:

*1) Requirement of Apln for angioblast migration*:

*The authors claim that not only Ela but also Apln guides angioblasts to the midline in the Results section and in the Title. In*
Figure 2*, the data of* apln^mu267-/-^
*clearly show that Apln is not required for angioblast migration. However, the authors point to the requirement of Apln using the embryos of the* ela^-/-^
*and* apln^-/-^. *On the contrary, they state the impact of Ela (instead of Apln)-Aplnr signaling in the Impact Statement. This interpretation might puzzle the readers. Therefore, they need to re-interpret the data and re-write the manuscript including the Title*.

We have clarified the inconsistencies, and re-written the manuscript accordingly.

*2) Vegf or Vegfa*:

*The authors use "Vegf" and "Vegfa" in the Abstract and main text. This inconsistency should be corrected not to mislead the readers. Are other* vegf *members except Vegfa are thought to regulate angioblast migration in the previous papers? According to their results, the conclusion might be that Vegfa-Vegfr2 signaling is required for angioblast migration toward the midline. Therefore, they should rewrite the manuscript by keeping the consistency when they use the similar words*.

We now always specify Vegfa, and have re-written the manuscript accordingly.

*Reviewer #3*:

*Helker and colleagues provide evidence for new roles of previously identified small peptides Apelin and Elabela in guiding angioblasts towards the embryonic midline during zebrafish embryogenesis. Previous studies implicated Vascular endothelial growth factor in convergence of angioblasts towards the embryonic midline, where they coalesce to build the first embryonic vessels. The authors first present evidence from zebrafish against the requirement for Vegf in angioblast migration towards the midline. Rather, they proposed, Apelin and Elabela, have partially redundant function to guide angioblasts to the midline by acting through the Apelin receptors. Apelin, Apelin GPCR Receptor axis has been previously implicated in cardiac progenitor cell convergence and cell fate specification, by Scott et al., and Zeng et al., in 2007. Elabela acting through Apelin receptors has been shown to be required for gastrulation cell movements and both heart and endoderm development by*
[4]
*and*
[26]*. However, neither Apelin nor Elabela have been proposed to affect gastrulation cell movements acting as guidance molecules, with Elabela being considered a motogen. The authors employ morpholino oligonucleotides and mutations to show that Apelin and Elabela expressed in the notochord have partially redundant function in angioblast migration. Moreover,* apelin receptors a *and* b *expressed in angioblasts also have redundant function in this process. Therefore the proposed role of Apelin/Elabela/Apelin receptor signaling to guide angioblasts towards the embryonic midline would be a significant advance. However, the evidence in support of these conclusions are not compelling. There are also problems with the results and their interpretations. These problems should be addressed before the manuscript becomes suitable for publication*.

*The most novel conclusion of this work is that Elabela/Apelin/Apelin receptor axis has an instructive role in angioblast migration, by guiding them to the midline. The key line of evidence is that when cells ectopically expressing Ela were present in Ela mutant: "We found that individual Ela overexpressing cell were indeed capable of attracting angioblast migration to ectopic locations" […] our data therefore demonstrate, that Ela acts as a chemoattractant for angioblasts during midline migration"*.

*The presented data is not compelling. The authors observed occasional overlap between angioblasts and these Ela expressing cells. An alternative interpretation is that Ela expressed in ectopic location increased motility of angioblasts and some of them wondered in ectopic locations*.

As described above, we have now added a new figure, in which distances between angioblast and control or *ela* expressing cells were measured. This analysis now shows a clear attractant role of Ela. By using *noto* or *ta* mutants, we have abolished *ela* and *apln* expression in the midline and could therefore score all angioblasts and not only those at very ectopic locations.

*The conclusion about Vegf signaling not being required in angioblast migration is based on the observation that "neither Vegf ligand depletion (*vegfaa *MO) nor a null mutation in the gene encoding the VEGFR2 ortholog (*kdrl*) interfered with angioblast migration in zebrafish embryos." However, there is no data showing the degree of loss-of-function for morpholino or for* kdrl *mutant*.

As described above this has been added, in Figure 1—figure supplement 1.

*Likewise, when the authors inhibit Shh signaling using cyclopamine, no quantitative data is presented documenting the degree of Shh signaling inhibition. Reduced expression of* vegf *is shown by in situ hybridization, but this is not a quantitative results. Such body of evidence is not sufficient to support the authors' conclusion regarding*:

The role of Shh (and cyclopamine treatment) on regulating Vegfaa has been published previously (with all controls). We regret that we have not expressed that well enough in the previous manuscript and have changed the wording. Additional experiments (chemical inhibtion of Vegfr2) have been added.

*Does mutation in* apelin *gene affect* elabela *expression and vice versa*?

*Do mutations in* kdrl *affect phenotypes of* apelin*,* elabela *or* apelin receptor *single mutants*?

*The authors conclude that simultaneous knockdown of* aplnra *and* b *affects angioblast cell movement but not specification, based on morpholino interference. Whereas previously published morpholinos are used, appropriate experiments showing efficiency and specificity should be presented. Given that mutations for these receptors are available, they should be preferably employed*.

We have by now generated mutant in the *aplnr* genes and were now able to analyze also double deficient embryos. Our mutant analysis confirms our morpholino generated data.

*Similar concerns are regarding* apelin *morpholino and* apelin *mutant experiments. Does* apelin *mutation lead to nonsense mediated degradation of* apelin *mRNA*?

*The authors conclude that angioblast migrate normally in* apelin *mutants, however, in*
Figure 2
*it appears that ca. 30% of* apelin *mutants showed mild angioblast migration phenotype. This along with a strong enhancement of the phenotype of* elabela *mutant indicates that* apelin *is also required for angioblast migration*.

*In the Results the authors write that cardiomyocyte deficiency observed in* aplnr *deficient embryos could not be phenocopied by loss of* apelin *function. This is not correct; Scott et al., and Zeng et al. reported reduction of cardiomyocyte formation in* apelin *morphant embryos*.

Scott et al claim: "Midline expression of the Agtrl1b ligand Apln suggests that it may provide a signal for migration of the myocardial progenitors. However, injection of three independent *apln* MOs (designed against the 5'UTR and the splice donor site in intron 1) caused severe defects in body axis elongation and morphogenesis (Figure 3), but did not recapitulate the *grn* myocardial phenotype, as significant myocardium was formed in the morphants (Figure 3). The MOs appear to be functional, as coinjection of the 5'UTR-targeted MO was sufficient to reverse the inhibition of cardiomyogenesis caused by *apln* RNA levels far exceeding those normally found in vivo (Figure 3). While exogenous *apln* can therefore recapitulate the *grn* phenotype, and Apln can act as a ligand to stimulate Agtrl1b activity, an *apln* mutant will be required to definitively test whether Apln is the in vivo signal for Agtrl1b during myocardial progenitor development.”

Zeng et al. are not as direct in their statements, but they claim: "However, the amount of cell death did not correlate with the severity of heart loss: we observed a level of cell death in *agtrl1b* morphants (where heart was strongly reduced) that was comparable to that in *apelin* morphants (with much milder reduction of the heart field, Figure 4 and data not shown).”

However, we have changed our wording to a less explicit: “Previous studies have identified a requirement for Apln receptors in myocard development in zebrafish (28; 33) and mice (3), which could not completely be phenocopied by Apln deficiency (3; 28; 33).”

[Editors' note: further revisions were requested prior to acceptance, as described below.]

*1) The lack of a generalised developmental delay (control for somite number) should be stated explicitly in the text as well as in the Materials and methods, so that it can be ruled out as a possible explanation for a delay or failure of angioblasts to migrate to the midline in MO-injected or mutant embryos*.

We have changed the text accordingly and pointed this out.

*2) The new data supporting the assertion that Ela acts as a chemoattractant and not a motogen are still based on rather low numbers, and indeed the new data are still presented only as a supplementary figure (Figure 3–figure supplement 1). In the text description (Results, seventh paragraph), n numbers (cells and embryos) must be quoted in the text to support the percentage values, in addition to the information provided in the figure*.

*The chemoattractive effect of Ela appears to be an important conclusion of the manuscript, but is only based on this supplementary figure. If the data are not strong enough to be shown as a main figure, the conclusion should also be toned down: rather than stating that the data 'demonstrate' that Ela acts as a chemoattractant, wording such as 'the data support the interpretation that Ela is acting as a chemoattractant in this context, but further experiments and higher n numbers would be required to test this more rigorously' should be used. In addition, use of the word 'chemoattractant' should be treated with similar caution throughout the manuscript*.

We had discussed already previously, whether we should include this figure into the main manuscript figures and have now added it as Figure 4. Figure 4 then has been relabeled to become Figure 5. We have also toned the wording down a little bit (at the position indicated and in the final summary). However, we believe our data not to be based on low numbers as we repeated the same experiment in the 2 notochord mutants and thereby generated separate data sets, each of them alone was statistically significant, so theoretically either of them would have already been sufficient.

*3)*
Figure 4
*and description and Results. Again, n numbers should be stated along with the percentages here, in the text as well as on the figure*.

We have changed the manuscript in the Results section accordingly. They were already indicated in the figure and stated in the figure legend in b.

*4) What do the error bars represent in*
Figure 4?

Now Figure 5. They actually represent the SEM, however we did not phrase this clearly. We have changed the manuscript accordingly and added the excel sheet used for calculation as a source file. We have referred to these source data in the figure legend of Figure 5.

As an example: transplants of ctr. MO in Wt was analyzed in 7 embryos (with totally 39 cells transplanted), of these in 5 embryos 100% of the transplanted cells migrated to the midline, in 1 embryo 75%, and in one embryo 80%, resulting in an average percentage of 93.57%.

Alternatively, it would also have been possible to calculate total numbers (i.e. 2 cells out of 39 cells did not migrate to the midline), but we wanted to reflect that there was not much variation between the individual embryos analyzed.